# Model-Based Enterprise Approach in the Product Lifecycle Management: State-of-the-Art and Future Research Directions

**Angelo Corallo** 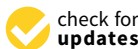, **Vito Del Vecchio ***, **Marianna Lezzi** and **Angela Luperto**

Department of Engineering for Innovation, University of Salento, 73100 Lecce, Italy;
angelo.corallo@unisalento.it (A.C.); marianna.lezzi@unisalento.it (M.L.); angela.luperto@unisalento.it (A.L.)
* Correspondence: vito.delvecchio@unisalento.it

**Abstract:** Innovations in product development and process technologies represent a promising strategy to face the increasing competitiveness of modern markets in the global economy. Also, customer requirements become more and more specific and the complexity of products is still increasing. Industries need to adopt effective solutions during the product development process and to support, for sustainable purposes, all the phases of the product lifecycle. Advanced model-based solutions emerge for digitally supporting these industrial needs. In this context, a Model-Based Enterprise (MBE) represents an organization that adopts modeling technologies, such as Model-Based Definition (MBD) solutions, to integrate and manage both technical and business processes related to product design, production, support, and retirement. Past research discusses the model-based approaches focusing on technical product development, mainly referring to the design and the manufacturing phases. A wide perspective from the other phases of lifecycle seems to lack in the literature. By carrying out a systematic literature review, this research aims to investigate the relationships between the MBE approaches and the product lifecycle phases and to identify potential benefits and challenges. It enhances the academic knowledge domain by also shedding light on potential future research directions.

**Keywords:** model-based enterprise; model-based definition; product lifecycle management; manufacturing; systematic literature review

## 1. Introduction

To be competitive in the modern markets and in the global economy, where the complexity of products is still increasing, the industries need to adopt effective solutions during the product development process and to support, for sustainable purposes, all the phases of product lifecycle [1,2]. The customer requirements are becoming more and more specific and the market demand tends to be very dynamic and also unpredictable in such cases. On the other hand, industrial organizations address new forms of product development, including multidisciplinary, cooperation, and co-design. Traditional approaches limit and challenge the businesses and the execution of their development activities, affecting the management of the product lifecycle phases [3]. Low-cost manufacturing of high-quality products also remains an essential part of the current economy, and technological advances made it possible [4]. Many product development processes are no longer serial step-by-step processes and, moreover, the design and the engineering activities are generally a waterfall process where engineers work independently and in parallel. For all these reasons, it is critical to have a single definitive data source to ensure data accuracy, consistency and to better manage the product lifecycle [5]. Advanced model-based solutions, including both technologies and methodologies, emerge for supporting digitally the development of a product and all its lifecycle phases, and to create a digital representation of real business processes and products. The transition from traditional product development practices (e.g., the use of 2D drawings) to digital drivers based on product models represents a

sustainable solution to: (i) optimize the product development process [6]; (ii) better manage the product lifecycle in the beginning-of-life, middle-of-life and end-of-life [7]; (iii) improve the communication between both technical and less-technical teams [8]; (iv) reduce the development time and costs and minimize the risk of non-compliance [9]. In particular, the paper faces the sustainability concept in relation to all the phases of the product lifecycle (e.g., sustainability-by-design, green manufacturing, and sustainable product disposal). The automotive and aerospace sectors are the first industries that have adopted these solutions. For example, Boeing took the lead in adopting MBD technology in the R&D (Research and Development) of Boeing 787 aircraft, achieving collaboration between the design and manufacturing partners [10]. In these contexts, the wide adoption of 3D models is common as the authoritative source of geometrical data. 2D drawings are still used (e.g., in manufacturing plants, technical documentation) but are directly generated from 3D models [11].

A Model-Based Enterprise (MBE) represents an organization that applies modeling and simulation technologies to integrate and manage all of its technical and business processes related to product design, production, support, and retirement [12]. By adopting product and process models for defining, executing, controlling, and managing all the enterprise processes, and by applying science-based simulation and analysis tools to optimize processes at each step of the product lifecycle, it is possible to reduce the time and the cost of product innovation, development, production, and support [13]. It is expected that the configuration of a model-based enterprise could potentially reduce the costs by 50% and the time to market by 45% if compared with common traditional practices [9].

A key component of the MBE approach is represented by the Model-Based Definition (MBD) [14] defined as the practice of using 3D CAD (Computer-Aided Design) models to mathematically describe the product or component specifications, including Product Manufacturing Information (PMI), annotations and other technical attributes [15]. The MBD is a part of a new strategy of product lifecycle management based on CAD models transition from simple gatherers of geometrical data to comprehensive sources of information for the overall product lifecycle [11]. A direct consequence of this shift is represented by the evolution of both businesses and computer-aided tools because of the usage of annotated 3D models that serve as the single source for all technical product information, also eliminating the need for 2D engineering drawings. Until recently, most engineering and manufacturing activities relied on hardcopy and/or digital documents to transfer engineering data and to lead the manufacturing processes [3]. Conversely, by enabling an integrated and collaborative environment based on 3D product definition details that are shared across the enterprise, a rapid, seamless, and affordable deployment of the product is ensured from concept to disposal. While the MBD has been gaining popularity in engineering and manufacturing environments, several questions remain unsolved regarding the full definition of MBD models. Standards such as ASME Y14.41 and ISO 16792 exist to document how a model should be defined with annotations. These standards also help in understanding how to interpret the data within the model. However, they do not document the required amount of information that the model must contain. It is important to understand what information needs to be communicated when considering moving from 2D drawings to 3D CAD models, so that engineers can efficiently perform their tasks [8].

Furthermore, the management of product information is a key activity that interests both technologies and methodologies and that affects all the lifecycle phases (i.e., plan, design, build, support, dispose) [16]. The adoption of MBE practices is becoming a reality in industry, as highlighted by the increasing number of companies that are moving towards model-based environments [14].

In this context, this research aims to investigate the relationships of model-based enterprise approaches, including model-based definition practices, with all phases of the product lifecycle. In particular, the focus is on understanding how MBE and MBD support, benefit, and challenge the organizations during the lifecycle phases. Previous research faces these topics mainly for the management of product information during the

technical development activities such as the design phase [17–20] and the manufacturing phase [3,20,21]. These links appear much aligned with the concept of PDM (Product Data Management) that includes systems for the handling of data throughout the whole design, engineering, and development process, also considering the control of workflows [22]. An interesting wide perspective from the side of the other phases of the product lifecycle seems to lack in the literature and this research wants to confirm or reject this statement. For this reason, the study analyzes the relationships between model-based enterprise approaches and the other phases of Product Lifecycle Management (PLM), crossing the boundaries of PDM. To achieve this goal, a systematic literature review has been carried out to build a knowledge base of reference in the domain of model-based enterprise and model-based definition.

The remaining of the paper is structured as follows. The next session details the research methodology adopted in this paper from both the strategy design and its application perspective. Following, the results of the literature review are presented and structured to have a comprehensive view of the potential existing relationships between MBE/MBD and the product lifecycle phases. The last sessions discuss the final remarks, including implications and limitations.

## 2. Materials and Methods

This study adopts the systematic literature review (SLR) approach [23] to investigate the relationship between the model-based enterprise (and thus model-based definition) approach and Product Lifecycle Management. For this reason, the study focuses on the literature analysis of papers in which explicit reference is made to the entire product lifecycle or one of its main phases (such as plan, design, build, support, and dispose [16]).

To achieve this goal, a systematic literature review process, based on keywords and search terms used through a replicable and defined search strategy [24–28], was adopted. Specifically, the steps underlying this study are described below:

1.  Definition: In the definition phase, the topic and search strategy are defined, as well as the scientific databases used. In addition, the search for material is carried out in the field of research under consideration through the identification and use of keywords. Therefore, search strings have to be defined.
2.  Execution: The execution phase is carried out as defined in the previous phase. In this step, papers are searched within the reference databases, both in relation to their relevance and by applying inclusion/exclusion criteria. Once the relevant papers have been identified, they are selected and extracted to perform the next stage of analysis.
3.  Analysis: In this phase, a descriptive and content analysis of the selected papers is carried out. In particular, the y are aggregated according to the areas of analysis, useful for achieving the defined research objective, and analyzed systematically.
4.  Evaluation: The last step consists of the comparative evaluation of the papers based on the considered areas of analysis.

A schematic view of the steps involved in carrying out this systematic literature review is shown in Figure 1.

### 2.1. Definition and Execution Phases

The collection of papers started by defining a search string for the major scientific databases, such as Scopus (www.scopus.com accessed on 30 November 2021) and Web of Science (www.webofknowledge.com accessed on 30 November 2021). The research took place until November 2021.

The search was first conducted using the following keywords: "MBE", "MBD", "Model-based Enterprise" and "Model-Based Definition", with the logical operator "OR" in the middle of each keyword. From the results obtained (29.511 document results with Scopus and 25.268 results with WoS), we realized that both the acronym MBE and MBD have different meanings unrelated to the topic in question. For example, some meanings of the MBD acronym are "multiple biodiversity" (environmental sustainability area), "mass

balance deviation" (in chemistry), "maximum bicondylar distance" (in social sciences), "mineral bone disorder" (in medicine) and "Minimal Brain Dysfunction" (environment and occupational health area); while some meanings of the MBE acronym are: "Molecular Beam Epitaxy" (physics, astronomy, chemistry, biochemistry, molecular biology and materials science area) and "minority business enterprise" (Construction & Building Technology area). For this reason, we have chosen to limit the search area to the keywords: "Model-Based Enterprise" and "Model-Based Definition" always with the OR operator in between.

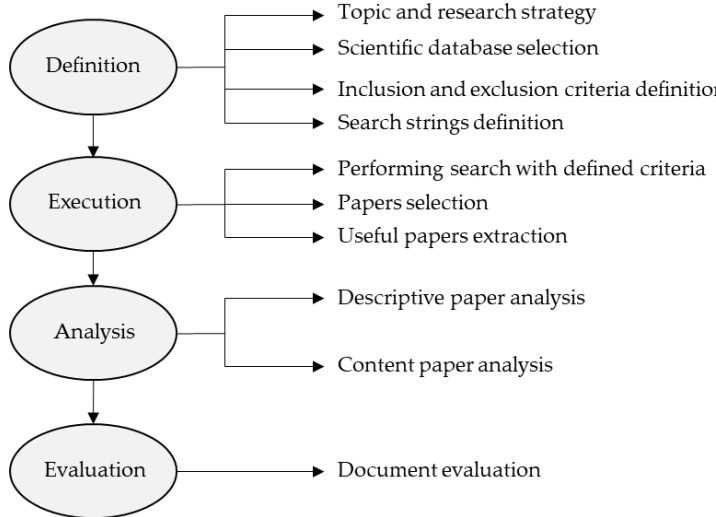

**Figure 1.** Outline of the systematic literature review.

In Scopus, the search was conducted in Article Title, Abstract, and Keywords, and returned 239 papers. On the other hand, the search in WoS was performed in Topic (including search title, abstract, keywords, and keywords plus), and returned 139 articles. As a result, a total of 378 scientific papers were identified in this first step.

Inclusion and Exclusion Criteria Definition

With the aim of selecting relevant papers in relation to the search objective, a filtering process of the articles was set up, defining inclusion and exclusion criteria (see Table 1).

First, the English language constraint was applied. This resulted in 209 articles for Scopus and 137 for Web of Science.

Secondly, the filter related to the research area of interest was used:

- Scopus: The subject areas considered are Engineering, Computer Science, Business, Management and Accounting, Social Sciences and Economics, Econometrics, and Finance. As a result, 195 articles were returned.
- WoS: The research areas considered are Business Economics, Computer Science, Engineering, Materials Science, Mechanics, Science Technology Other Topics. As a result, 123 papers were returned.

**Table 1.** Inclusion/Exclusion criteria.

| | Criteria | Description |
|---|---|---|
| Inclusion | Consideration of abstract | Documents containing an abstract focused on MBE and MBD are included |
| | Consideration of paper | Documents related to the topic of study are included |
| | Document type | All the types of documents present in the databases are considered |
| Exclusion | No English paper | Documents that are not written in English are excluded |
| | Unrelated area | Off-topic documents with respect to research areas of interest are excluded. |
| | Duplicate Documents | Documents repeated in different scientific databases are excluded. |

Finally, applying the last exclusion criterion (duplicate documents) resulted in 206 papers to be analyzed in the next review phase (see Table 2).

**Table 2.** Papers resulting from the application of filters.

| Papers | Scopus | WoS | Total |
|---|---|---|---|
| Found | 239 | 139 | 378 |
| In English | 209 | 137 | 346 |
| Related to the research area | 195 | 123 | 318 |
| Duplicated | | | 112 |
| Total | | | 206 |

### 2.2. Analysis Phase

In this phase, a matrix was first defined as a support to assess the actual contribution of the 206 selected papers. The matrix consisted of 17 columns in which the following information was collected: papers ID, research database, authors, title, source, paper type, year, paper purpose, paper confirmation after reading the abstract and after reading the article, focus, research area, research area subcategory, industry, contribution to PLM, implications for future research, and finally implications for practice and main results.

For each paper, this matrix was compiled for the first eight points, relating to general information. Subsequently, reading the abstract made it possible to exclude those papers that were clearly off-topic (out of 206 articles, 75 passed the selection). A further selection of the papers was made by reading their entire content; following this in-depth analysis, it emerged that the useful papers referring to the study domain were reduced to 19.

Therefore, a second assessment phase was carried out on these 19 papers, which led to the compilation of the remaining points of the matrix (points 9 to 17). An extract of the matrix with the most relevant information is available in Appendix A.

### 3. Papers Evaluation Phase

Before introducing the results of the literature review on MBE and MBD solutions, it is important to provide a concise definition of each phase of the product lifecycle to understand the related sub-activities and link the potential of model-based approaches. Considering the study of Grieves [29], the product lifecycle is composed of five main phases:

- Plan: the product model starts from the requirements analysis which is the first step in the development process. The requirements come either directly from the customer or indirectly from marketing, which analyses market needs;
- Design: starting from the requirements, the concept, and, subsequently, the prototype of the product is developed. Different alternative prototype options can be implemented that meet the same requirements with different functions and technologies;
- Build: when the product is completely defined, manufacturing determines how to build it. Different issues are considered depending on whether or not there is a suitable plant or machinery to make the product in question;
- Support: maintenance, sales, and distribution functions use product information to demonstrate product features and characteristics to the customers, and to understand whether they can meet their needs;
- Dispose: retirement, disposal, and recycling concepts close the product life cycle and product information is necessary for these activities to be carried out efficiently.

Considering that the research aims to investigate the relationship between the model-based enterprise (and therefore model-based definition) approach and product lifecycle management, two macro-areas of analysis are considered: (1) the role of MBE/MBD in relation to product lifecycle and (2) the benefits and challenges of using MBD technology in relation to the PLM. These macro-areas are detailed below.

*3.1. The Role of MBE/MBD in Relation to the Product Lifecycle*

3.1.1. MBE/MBD in the Entire Lifecycle

Some of the 19 selected papers address the issue of data management through MBD models in relation to the whole product lifecycle.

In particular, the survey conducted by Ruemler et al. [8] on the use of MBD in the industry shows an interest in the use of this technology, but also a difficulty in its adoption due to the lack of a single source for managing data and information across workflows.

Adamenko, Pluhnau and Nagarajah [21] and Pippenger [30] state that manufacturing companies should evolve from document-centered data and information management to an MBD approach so that all relevant information for each product lifecycle phase is obtained from a single source, eliminating the need for many models, protocols, and redundant documentation. This prevents important data and information from being lost along the different phases of the product life cycle. Using the MBD approach, the necessary information and data are properly stored in CAD files, in annotated form, according to the process and product lifecycle phases considered [30]. In this way, the geometry of the product part can be simplified according to its role, process, and function thus simplifying data management, ensuring the data exchange across and inside the company, and protecting its know-how.

Hartman, Rosche, and Fischer [31] differentiate the type of CAD files to be used in the following formats: native CAD formats, derivative data formats, lightweight collaborative files, and neutral files. This distinction is important because the product representations used across the lifecycle change with respect to the desired functionality at each stage, and the exclusive use of one class of representation over another could limit the flexibility of design tools. On the other hand, one of the basic principles for product lifecycle management and model-based enterprise is that movements and transfers of information throughout the enterprise to enable effective decision-making can be accomplished by a high-fidelity digital product representation.

In this regard, the work of Rinos et al. [32] aims to eliminate counterproductive data formats used in industry (e.g., those based on 2D drawings) to optimize collaboration between different company departments. For this reason, they propose a method that uses an MBD 3D PDF-based template to create a document in a lightweight data format that contains the necessary information for all product lifecycle processes.

In line with this view, Briggs et al. [33] highlight the need for a transition from production based on 2D drawings to one in which all stages depend on model-based definition data. This transition can enable integration between the tools and related MBD outputs of one organization's engineering and those of other organizations. To this end, it is important to make product definition data and MBD accessible, viewable, and usable by users at all stages of the product lifecycle.

Yang et al. [2] state that the quality of the MBD model plays a key role in achieving model-based enterprise. In particular, they promote the use of flexible tools and standards to avoid misunderstandings between the stakeholders involved in the use of these models at different phases along the product lifecycle. If inconsistencies arise in product data and MBD models, the result could be increased production costs, an extended production cycle, and failure to achieve MBE.

From the research conducted by Wardhani et al. [34] and Trainer et al. [35], it is possible to recognize that there are gaps in standards, as well as in tools, that do not allow the MBD implementation and therefore the industry progress towards MBE. In particular, they focus on the STEP (Standard for Exchange of Product Model Data) AP242 standard and discuss the possibility of fully defining product data in the product lifecycle and enabling collaboration between different CAD systems. However, issues related to the different proprietary data formats are highlighted; the current MBD approach still does not support the storage of the information needed in the different stages of the product lifecycle [34].

Similarly, Hedberg et al. [36] argue that standards-based information integration is not feasible today. In fact, their research, which aims to test the ability of consensus-based

data standards to integrate product lifecycle stages through the implementation of a small model-based enterprise, shows that popular data standards used in the industry do not support automatic data alignment without significant human intervention.

On the other hand, Goher, Shehab, and Al-Ashaab [37] believe that issues in MBD development and implementation can be divided into three macro-categories: (1) technical issues (such as knowledge of product definition elements and information flow, and use of standards); (2) management issues (in terms of changing from conventional drawings to MBD models); and (3) certification issues (i.e., design data should have the characteristics for maintaining availability, accessibility, integrity, quality and security throughout the product life cycle).

Furthermore, Alemanni, Destefanis, and Vezzetti [11] claim that companies often lack a comprehensive strategy and appropriate methods to support the development of MBD. Therefore, they propose a unified and objective approach based on the QFD (Quality Function Deployment) model to define the MBD. This is a common methodology to structure data into reusable and unified forms within 3D models at all product lifecycle phases.

Finally, Zhang et al. [10] state that MBD-based integrated data management is a key technology to enable model-driven dynamic synchronization of activities involved in the production of complicated and customized products (C&CP). This technology achieves efficient collaboration between different business activities, because the MBD dataset can fully describe both geometric and non-geometric information related to different parts of the product, preserving data consistency and connectivity during its lifecycle.

### 3.1.2. MBE/MBD in the Design Phase

Other analyzed papers focus on the importance of introducing MBD practices for enhancing the design phase of the product lifecycle.

In particular, to improve the design efficiency, Huang et al. [17] propose a 3D process design method based on MBD technology that aims to provide a theoretical basis for the realization of the 3D process design of complex systems. This method overcomes the difficulties related to: (i) the heterogeneity of data sources in the design phase; (ii) the non-uniformity of information between different company departments; (iii) the redundancy of data throughout the design and production process.

MBD technology renders technical information in a three-dimensional environment, creating an MBD dataset that can fully describe the product [18]. For this reason, Yang et al. [18] propose the MBD attributes method, in which MBD attribute models are created by combining the various MBD attributes associated with different product types. In this way, the designer can directly select the attribute values, thus reducing his workload and improving the integrity and accuracy of the MBD dataset and consequently also the design process.

Duan, Shen, and Liu [19] also believe that introducing MBD in the design phase could bring valuable benefits to the company. In particular, the authors focus on the design part inherent in component assembly and investigate a solution to facilitate the MBD integration in relative position accuracy (RPA) measurement in order to make the products' parts or components compliant with the design specifications.

Finally, Zhu et al. [20] focus on the implementation of MBD technology in advanced design. Specifically, they state that the MBD design model should be characterized by three main entities, such as: the definition of the design model for each part of the product; the use of 3D annotations; and the explication of product attributes.

### 3.1.3. MBE/MBD in the Build Phase

Some of the analyzed papers address the adoption of the MBD model for supporting the manufacturing (or building) phase of the product lifecycle.

In particular, Liu, Duan, and Liu [38] state that MBD model-based inspection plays an important role in manufacturing processes because the information can be integrated into the 3D model providing a unified product definition. Therefore, they propose a concept of

integrated model-based inspection to promote the integration of design and manufacturing and to improve manufacturing efficiency and quality control capability.

Likewise, Hedberg et al. [3] state that to realize the vision of MBE, a single "digital thread" must be created. The digital thread would enable real-time design and analysis, collaborative process flow development, automated artifact creation, and full process traceability in a seamless real-time collaborative development between stakeholders. To achieve this goal, the authors emphasize the importance of filling the lack of standards for defining PMI (Product Manufacturing Information) so that data can be interpreted and presented consistently by different engineering and manufacturing operations.

On the other hand, Adamenko, Pluhnau, and Nagarajah [21] state that many manufacturing processes are still centered around documents or drawings. Moreover, organizations often use the same drawings for as many departments as possible in order to avoid redundant models in PLM systems. The negative effects of this trend are several: (i) drawings are overloaded with information; (ii) manufacturing models contain information that is not needed for that process; and (iii) manufacturing engineers must spend additional time finding the information they need.

Finally, the study conducted by Zhu et al. [20] focuses on the MBD technology implementation in complex manufacturing systems as a new form of collaboration. The authors state that it is necessary to create an MBD process model and not rely only on the MBD model of the design phase. In fact, the MBD design model does not consider intermediate manufacturing states of parts, but only provides process information. Therefore, an integration of both the design and manufacturing systems is required to efficiently use MBD technology.

### 3.2. Benefits and Challenges of Using MBD Technology in Relation to PLM

This section aims to gather the benefits and challenges that emerged from the literature review, regarding the use of MBD technology in relation to PLM.

According to [17,19], the use of MBD technology in design and manufacturing processes has improved data integration and made the workload associated with design, manufacturing, and assembly personnel efficient, thus contributing to improved process performance.

Moreover, Zhang et al. [10] and Yang et al. [18] state that product lifecycle management can be improved in terms of data consistency and connectivity through the implementation of MBD-based 3D design. MBD-based 3D design technology uses the MBD as a single data source by defining 3D design information, 3D manufacturing information, and product management information in the 3D digital model of the product. Therefore, the MBD can be adopted across the entire product demand model (design, process, manufacturing, and service model) to support the coordination of the product tooling, manufacturing, assembly, and maintenance process by setting the product design parameter and breaking down the barriers between design, manufacturing and operation and maintenance information.

On the other hand, Ruemler et al. [8] and Alemanni, Destefanis, and Vezzetti [11] argue that model-based definition is a strategy to move from two-dimensional (2D) paper drawings to three-dimensional (3D) computer-aided design (CAD) models, where the model contains all the information so that drawings may no longer be needed. This results in shorter time-to-market, more efficient processes, and better product quality. Product models are also crucial to achieve interoperability between applications, people, and companies, as well as data exchange. However, the same authors claim that MBD development today mainly concerns data structures that need to be in reusable forms and unified within native three-dimensional CAD models. For this reason, it will be necessary to propose a global strategy and appropriate methods to support MBD development and define new standards and common practices to create a common language for modeling and data management.

The study conducted by Hedberg et al. [3] on the comparison of model-based versus design-based processes found that model-based processes result in a cycle time reduction

of 74.8% compared to design-based processes; however, both present challenges related to the fulfillment of the design and manufacturing phases of the product life cycle.

According to Pippenger [30], moving to a full MBD environment raises a number of challenges regarding data accessibility and visualization, data content, data presentation, data management, data security, and data retention.

Furthermore, to integrate the concept of model-based enterprise into the industrial world, the MBE strategy has to ensure interoperability of model-based data from the design and manufacturing stages through to the support stage in the supply chain [35]. However, several barriers to the interoperability of model-based data have been identified [35]. In particular, the two-dimensional (2D) drawing is considered the legal record of master data compared to the three-dimensional (3D) model; moreover, in the context of automation, many application programming interfaces do not adequately support the reading and writing of standards-based Product Manufacturing Information. Finally, easy data exchange through standards-based implementations threatens to disrupt the business model of major product lifecycle management tools.

In general, a robust MBE inherently depends on the easiness of data transformation, which is significantly enhanced by the collaborative capabilities of the modeling tools used to create data and the standards used to exchange that data. In fact, the application of appropriated standards ensures that data flows seamlessly throughout the product lifecycle and allows for the reuse of data in the most appropriate formats for collaboration and visualization [31].

Table 3 summarizes the most significant benefits and challenges that emerged from the literature study regarding the use of MBD technology in relation to PLM. These benefits and challenges are listed in no particular order of priority and are aggregated for the different research papers analyzed.

**Table 3.** Benefits and challenges of MBD technology related to PLM.

| Benefits | Challenges |
| --- | --- |
| <ul><li>Cycle time reduction of 74.8% compared to design-based processes [3]</li><li>Improved data consistency and connectivity [10,18]</li><li>Supporting the coordination of product tooling, manufacturing, assembly, and maintenance processes [10,18]</li><li>Improved process performance [8,11,17,19]</li><li>Shorter time-to-market [8,11]</li><li>Better product quality [8,11]</li><li>Interoperability between applications, people, and companies [8,11]</li><li>Improved data integration [17,19]</li><li>Efficiency in the workload associated with design, manufacturing, and assembly personnel [17,19]</li></ul> | <ul><li>Need for a comprehensive strategy and appropriate methods to support the development of MBD [8,11]</li><li>Definition of new standards and common practices to create a common language for modeling and data management [8,11,31]</li><li>Data accessibility and visualization, data content, data presentation, data management, data security, and data retention [30]</li><li>Interoperability of model-based data from the design and manufacturing stages through to the support stage in the supply chain [35]</li></ul> |

## 4. Discussion

Before discussing the results obtained from the systematic literature review, it is interesting to understand the reasons why the analysis of the model-based enterprise has been carried out from the product lifecycle perspective. Several previous researchers [2,11,13] have recognized an intrinsic relationship between the MBE and product lifecycle by remarking the benefits and challenges of introducing model-based technologies for managing product data and information, and by highlighting the limitations of traditional approaches based on 2D drawings or incomplete product 3D models. Moreover, a direct relationship between MBE approaches and the PDM seems to exist because of the relevant contributions referred to the technical development phases of the product data management (i.e., design and build). The broad lifecycle perspective is under-researched. The lifecycle phases are generally conceptualized in a cyclic and sequential model. By connecting and integrating

each activity, it can be helpful to ensure concurrent and simultaneous engineering reducing the development time and improving the quality [39,40]. Each phase produces valuable information and uses the information deriving from the other phases, generating an information backbone [16]. Thus, it is interesting in recognizing how model-based approaches can benefit all phases of PLM (i.e., plan, design, build, support, and dispose).

The results obtained from the literature review allowed for the qualitative clustering of the analyzed papers that address the topic of model-based enterprise, including model-based definition, in relation to the different lifecycle phases (see Figure 2).

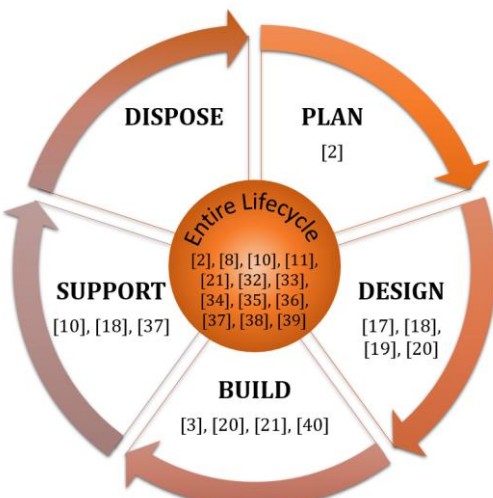

**Figure 2.** Clustering of MBE research in product lifecycle phases [2,3,8,10,11,17–21,32–40].

Most research papers focused on the adoption of model-based enterprise approach, and model-based definition practices, in relation to the design phase [17–20] and the manufacturing (a.k.a. build) phase [3,20,21,38]. Accordingly, these studies discussed how model-based solutions can support designers in managing product design data and in completely defining the 3D product model, also resulting in the reduction of the development costs as one of the most important benefits. On the other hand, MBE and MBD approaches support manufacturers in quickly retrieving the right product information during the production activities, enhancing the communication between departmental functions, and making available a unique source of product information in a shared data repository.

With respect to the other analyzed papers [2,8,10,11,21,30–37], the importance of model-based solutions is discussed in broad terms, considering the whole product lifecycle and addressing different benefits such as the possibility to improve the integration between the department functions of a company, and different challenges such as the need to use standards for ensuring data consistency.

However, some few papers pose to give attention to model-based enterprise by also considering other elements related to the other phases of the product lifecycle. For instance, Yang et al. [2] considered the MBD model for ensuring the quality of requirements and facilitating communication between different stakeholders. This kind of activity is related also to the planning phase of the product lifecycle.

Furthermore, Zhang et al. [10] and Yang et al. [18] remarked on the usefulness of adopting MBD practices to facilitate the coordination between the design, manufacturing, and service functions, as well as to boost the information retrieval for sustaining not only the design and manufacturing processes, but also the maintenance operations during the support phase of the product lifecycle. In the same way, also Trainer et al. [35] argued that MBE strategies represent an important driver for the support phase of the product lifecycle and for keeping up the supply chain management.

Therefore, on the basis of the results of this systematic literature review, it can be inferred that a direct relationship between model-based approaches and the product lifecycle mainly arises with respect to the design and the manufacturing phases. However,

some reflections can be made on the planning and support phases of the product lifecycle, while the disposal phase seems not to have been taken into consideration. Moreover, other multiple references addressed this topic in a broad meaning of lifecycle by also focusing on the data management process. Regarding the most important benefits of model-based solutions, they are related to the improvement of business performance in terms of time reduction for product design and manufacturing, and the enhancement of the coordination and collaboration of different departmental functions. However, important challenges have to be considered for ensuring some data issues like security, retention, and interoperability.

To fill the gaps in the literature and reduce the challenges seen in Table 3, the systematic literature review enabled the identification of future research directions for the use of MBD technology and consequently the implementation of MBE. These directions are detailed in relation to the phases of the product lifecycle (see Table 4).

**Table 4.** Future research directions of MBE in Product Lifecycle Management.

| Lifecycle Phases | Future Research Directions |
|---|---|
| Entire Lifecycle | • Define and provide robust guidelines useful to support MBD development<br>• Provide a framework about suitable standards for MBE implementation<br>• Encourage studies on empirical approaches, also including case studies of industrial organizations |
| Plan | • Investigate how the adoption of the MBE approach can support the engagement of stakeholders<br>• Examine in-depth the concept of MBD model quality and its impact in terms of requirements collection |
| Design | • Address the issue of data sharing in order to face data interoperability, consistency, and security |
| Build | • Analyze the management of product information considering the integration between the design and manufacturing phases |
| Support | • Investigate in-depth MBE strategies for supporting product maintenance operations |
| Dispose | • Understand how model-based solutions can benefit the end-of-life of products in terms of product retirement, recycling, or reuse |

## 5. Conclusions

Model-based solutions emerge in industrial environments for supporting the development teams during the product development phases of its lifecycle, as well as to sustain the other less-technical phases. While the Model-Based Enterprise (MBE) represents the organizational approach of applying modeling and simulation technologies for improving the performance of the development process, the Model-Based Definition (MBD) can be seen as an enabling digital driver based on the use of 3D CAD models to fully define the product, including all the related information.

Past research focused on model-based approaches by mainly considering the technical development phases, such as the design and the manufacturing phases of the product lifecycle. This research aimed to review the reference literature in order (i) to identify the potential relationships of MBE and MBD with the other phases of the product lifecycle (i.e., plan, support, and dispose), (ii) to collect the most recognized benefits and challenges that are related to the introduction of these technologies and practices in industries, and (iii) to identify potential future research directions.

The consideration of MBE and MBD practices, methodologies, and technologies is seen as a sustainable strategy to support organizations in the product development process as well as all phases of their lifecycle. Both the economic and business aspects of sustainability are faced. Indeed, the most important benefits generated by model-based solutions are related to the reduction of product development time and costs and the management of product information during the product lifecycle as well as the organizational functions. Furthermore, model-based enterprises consider a stronger integration of all interested stakeholders during the various stages of the product lifecycle, as a means of social sustainability and inclusion. Also, they consider model-based solutions to manage product

information for sustainability-by-design and green purposes, as a means of environmental sustainability that aims to manage product retirement, recycling, or reuse. This means that the inclusion of sustainable practices during the product design affects the building phase for reducing the product non-compliance rate, and the disposal phase for foreseeing the product end-of-life.

From the academic side, this work contributes to enhancing the knowledge domain by offering a wide perspective of model-based approaches in relation to the entire product lifecycle, and by systematizing previous research in a framework that clarifies the most addressed lifecycle phases. From the industrial side, this work supports organizations in increasing the awareness about the potential of model-based approaches and on how these solutions can affect both technical and less-technical phases of the product lifecycle.

The adopted conceptual and qualitative analysis represents a limitation of this research because it is only based on the literature contributions. Indeed, an empirical approach is encouraged for future research by also including evidence from business organizations. However, this work can represent a useful starting point since it sheds light on different suggestions to investigate the relationships between model-based approaches and the product lifecycle.

**Author Contributions:** The authors take part to the research work providing the following contributions: (i) introduction, V.D.V. and A.C.; (ii) materials and methods, M.L. and A.L.; (iii) paper evaluation, M.L. and A.L.; (iv) discussion, V.D.V. and A.L.; (v) conclusions, V.D.V. and M.L.; (vi) writing–original draft preparation, V.D.V., M.L. and A.L.; (vii) writing–review and editing, A.C., V.D.V., M.L. and A.L.; (viii) supervision, A.C. All authors have read and agreed to the published version of the manuscript.

**Funding:** This research received no external funding.

**Conflicts of Interest:** The authors declare no conflict of interest.

## Appendix A

**Table A1.** Details of the results obtained from the systematic literature review.

| ID | Authors | Title | Year | Focus | Contribution to the Product Lifecycle | References |
|---|---|---|---|---|---|---|
| 1 | Briggs et al. | Model-Based Definition | 2010 | It considers the role of model-based definition in the product development lifecycle. Software currently available for three different companies is examined, and aspects of recent implementations are outlined to delineate an approach for formulating the business case for adoption. | • Entire lifecycle | [33] |
| 2 | Alemanni, Destefanis and Vezzetti | Model-based definition design in the product lifecycle management scenario | 2011 | It focuses on a method for supporting the MBD implementation by the use of the quality function deployment approach. Three scenarios in which industrial companies working in the PLM domain were studied to achieve a standardized MBD data structure. | • Entire lifecycle | [11] |
| 3 | Hartman, Rosche, and Fischer | A framework for evaluating collaborative product representations in product lifecycle workflows | 2012 | It describes current models of the product development process and the nature of collaborative data. A framework is presented for evaluating various collaborative product representations and data. | • Entire lifecycle | [31] |
| 4 | Pippenger | Three-dimensional model for manufacturing and inspection | 2013 | It assesses the value for organizations of moving towards a three-dimensional model definition of their products. It examines the needs, risks, and benefits of this environment and the future models for manufacturing. | • Entire lifecycle | [30] |

**Table A1.** *Cont.*

| ID | Authors | Title | Year | Focus | Contribution to the Product Lifecycle | References |
|---|---|---|---|---|---|---|
| 5 | Trainer et al. | Gaps analysis of integrating product design, manufacturing, and quality data in the supply chain using a model-based definition | 2016 | It investigates three concepts: the ability to utilize a STEP AP242 model with embedded PMI for CAD-to-CAM and CAD-to-CMM data exchange; the gaps in tools, standards, and processes that inhibit industry's ability to cost-effectively achieve model-based-data interoperability in the pursuit of the MBE vision; the interaction between CAD and CMM processes. | • Entire lifecycle<br>• Support | [35] |
| 6 | Zhu et al. | Implementations of Model-Based Definition and Product Lifecycle Management Technologies: a Case Study in Chinese Aeronautical Industry | 2016 | It describes and analyzes the Model-Based Definition (MBD) and Product Lifecycle Management (PLM) technologies and their related Computer-Aided X applications in order to enable the implementation of an integrated design and manufacturing system in the aeronautical industry. | • Design<br>• Build | [20] |
| 7 | Hedberg et al. | Testing the digital thread in support of model-based manufacturing and inspection | 2016 | It documents the strengths and weaknesses in the current industry strategies for implementing MBE. It identifies gaps in the transition and/or exchange of data between various manufacturing processes. It presents measured results from a study of model-based processes compared to drawing-based processes. | • Build | [3] |
| 8 | Ruemler et al. | Promoting model-based definition to establish a complete product definition | 2017 | It carries out a survey to analyze the adoption of three-dimensional models in industries and to help in identifying the needed information to move from drawings to models. | • Entire lifecycle | [8] |
| 9 | Yang et al. | MBD attributes template method of aeronautical products | 2017 | It proposes to use the MBD attributes template method to solve the generation problem of attributes information in the MBD dataset. | • Design<br>• Support | [18] |
| 10 | Wardhani et al. | An approach to complete product definition using step in cloud manufacturing | 2018 | A consolidated approach is provided to complete the product definition based on the STEP AP242 neutral data format using the general notes data structure. A case study demonstrates the validity of this solution. | • Entire lifecycle | [34] |
| 11 | Zhang et al. | A model-driven dynamic synchronization mechanism of lifecycle business activity for complicated and customized products | 2019 | It proposes a framework for a dynamic business lifecycle synchronization mechanism for C&CP (complicated and customized products). It allows for efficient coordination of C&CP design, manufacturing, and O&M (operation & maintenance) activities. | • Entire lifecycle<br>• Support | [10] |
| 12 | Liu, Duan, and Liu | A framework for model-based integrated inspection | 2019 | A framework for MBI is proposed to promote the integration among design, manufacturing, and inspection as well as the integration among procedures inside the inspection processes. The MBD model is taken as the unified data source and information throughout design, manufacturing, and inspection processes. | • Build | [38] |
| 13 | Duan, Shen, and Liu | An MBD based framework for relative position accuracy measurement in the digital assembly of large-scale component | 2019 | It analyses the ways to facilitate inspection planning and promote the integration by introducing the MBD into RPA (relative position accuracy) measurement. On the basis of a framework, a prototype system is developed and a case study of aircraft landing gear assembly is conducted. | • Design | [19] |
| 14 | Huang et al. | Research on the Three-Dimensional Process Design Method of Shipbuilding Based on MBD Technology | 2019 | It discusses the 3D digital model and the process design method of shipbuilding based on MBD technology. A theoretical basis for the implementation of 3D shipbuilding process design is provided. | • Design | [17] |
| 15 | Adamenko, Pluhnau and Nagarajah | Case study of model-based definition and mixed reality implementation in the product lifecycle | 2020 | It analyses how the product-relevant information is integrated into a 3D model and can be used at several stages of the product lifecycle. It aims to achieve a model-based product development. | • Entire lifecycle<br>• Build | [21] |
| 16 | Yang et al. | A knowledge-based system for quality analysis in model-based design | 2020 | It proposes a knowledge-based MBD part model quality analysis system and its implementation technologies to analyze and test the quality of the model from the perspective of different model-used stages. | • Entire lifecycle<br>• Plan | [2] |

**Table A1.** *Cont.*

| ID | Authors | Title | Year | Focus | Contribution to the Product Lifecycle | References |
|----|---------|-------|------|-------|--------------------------------------|------------|
| 17 | Goher, Shehab and Al-Ashaab | Model-Based Definition and Enterprise:State-of-the-art and future trends | 2020 | It aims to review the literature on Model-Based Definition (MBD) and Model-Based Enterprise (MBE) to recognize the main contributions towards the development and implementation of MBD and explore its various perspectives. | • Entire lifecycle | [37] |
| 18 | Rinos et al. | Implementation of model-based definition and product data management for the optimization of industrial collaboration and productivity | 2021 | It proposes a methodology that uses the capabilities of MBD technology along with the use of PDM software to refine the data sharing process and streamline the collaboration among different departments of a company, without being limited to the design and manufacturing of the product. | • Entire lifecycle | [32] |
| 19 | Hedberg et al. | Defining requirements for integrating information between design, manufacturing, and inspection | 2021 | An experiment was conducted to test selected open data standards' ability to integrate the lifecycle stages of engineering design, manufacturing, and quality assurance through the thorough implementation of a small-scale model-based enterprise. | • Entire lifecycle | [36] |

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
