# Peer review of "Model-Based Enterprise Approach in the Product Lifecycle Management: State-of-the-Art and Future Research Directions"

_sustainability, doi:10.3390/su14031370_

Round 1

Reviewer 1 Report

An interesting paper and ideas may be useful, but there are a number of major concerns.

The initial question is does it fit with the journals scope?   There could be a stronger focus on sustainability upfront.

The research  aim needs attention… “research aims to investigate the adoption of MBE approach during the whole product lifecycle in 20 order to identify potential relationships with all the phases.”  Page 2 line 88.I  s the paper really about adoption? As you go further through the paper the aim changes ? Other aims throughout the paper different from this one.  In essence the paper develops a list of benefits and challenges.  It does not really address adoption which would actually be more of a contribution than the paper actually makes

The discussion and analysis need more depth and rigour.  At present it is somewhat repeating itself and very descriptive.

In relation to the order of presentation of table 3. It claims to present the most significant benefits and challenges.  What criteria was used to make these decisions?  How was it decided that these were the most significant.  What order are they presented in the table. 

 Section 4 the discussion goes back to the broader literature on Life Cycle.  This could fit  with the broader introduction and should be more concise and may help address the “within scope question”

The paper claims to “shed light”, there should be a stronger contribution than shedding light.  A clearer and more robust set of future research directions should be developed.  The paper very briefly links to future research.  In order to make a contribution this would need to be very substantial future research agenda.

The language could be somewhat more specific, formal in places. “in last years” page 1, line 10, page 1 line 27, nowadays.  It would benefit from a proofread by native English speakers.

Author Response

Dear Reviewer,

We thank you for the revisions, they are very useful to better clarify specific parts. We hope to have addressed all the reviews and we are ready to meet further requests.

Best Regards

The authors

Reviewer 2 Report

The aim of the paper is to analyse, based on a survey of the literature, the use of model based enterprise approaches and model based definition practices in product lifecycle management and to investigate benefits and challenges of adopting Model Based Enterprise approaches in the whole product lifecycle.

As the paper stands, although addressing the objective set for the research, it does not provide a breakthrough and would need improvements in the following directions:

1) the rationale and objectives of the research should be better clarified. What is the specific scientific issue that is addressed?

2) the list of references is limited to 37 entries. Make sure the scope of the work is entirely covered and the list of references is complete. Especially, he relationships of Product Data Model (PDM) and Product Information Model (PIM) with Model-Based Definition should be explicited. PDM/PIM should be covered to some extend. The same applies to Product-Service System (PSS) in the context of PDM.

3) Potential future research directions should be better discussed and strengthened as well as findings drawn from the literature analysis. The paper should go further in terms of drawing generalisations than a critical review of selected papers and the summary made in Table 3.

4) Although the English style is correct, a number of English spellings must be corrected throughout the paper.

Author Response

(The authors gave the same response as above.)

Round 2

Reviewer 1 Report

The paper is far more focused on achieving its aim and objectives and has moved toward generating ideas and avenues for future research.  The contribution of the paper is far clearer and overall better-rounded. 

Lines 45 to 49 need a citation.

Also it could benefit from another proof read and improve clarity/wording in certain areas. 

Author Response

Dear Reviewers,

We thank you for the revisions, they are very useful to better clarify specific parts. We hope to have addressed all the reviews and we are ready to meet further requests.

Below you can find a detailed answer (in green) to each comment.

Best Regards

The authors

Reviewer 2 Report

The paper has been significantly improved in terms of clarity of the objectives of the research, presentation and organisation, logical presentation of the argumentation and improvement of the English.

It now reads fluently and logically and additional material has been added to make it more comprehensive.

The remarks of the reviewers have been correctly addressed. The link with PDM has been explicited and a few sentences have been added to consider "sustainability".

  • the link to sustainability be a little bit more explained, both at the beginning of the paper and at the end
  • still a few English mistakes be corrected (for instances, do not start sentences with "In order to" but by "To ..." or some "s" missing at the 3rd person)

Author Response

(The authors gave the same response as above.)
